# Identifying Key Pathogens and Effective Control Agents for *Astragalus membranaceus* var. *mongholicus* Root Rot

**DOI:** 10.3390/jof11070544

**Published:** 2025-07-21

**Authors:** Bo Zhang, Bingyan Xia, Chunyan Wang, Ouli Xiao, Tielin Wang, Haoran Zhao, Xiaofeng Dai, Jieyin Chen, Yonggang Wang, Zhiqiang Kong

**Affiliations:** 1State Key Laboratory for Biology of Plant Diseases and Insect Pests, Institute of Plant Protection, Chinese Academy of Agricultural Sciences, Beijing 100193, China; zb981210@163.com (B.Z.); xiabingyan2001@163.com (B.X.); xiaoouli123@163.com (O.X.); zhaohr98@163.com (H.Z.); daixiaofeng_caas@126.com (X.D.); chenjieyin@caas.cn (J.C.); 2School of Life Science and Engineering, Lanzhou University of Technology, Lanzhou 730050, China; wangyg@lut.edu.cn; 3College of Agronomy, Qingdao Agricultural University, Qingdao 266109, China; wangcy@126.com; 4National Resource Center for Chinese Materia Medica, China Academy of Chinese Medical Sciences, Beijing 100700, China; wtl82@163.com; 5Beijing Advanced Innovation Center for Food Nutrition and Human Health, School of Light Industry and Technology, Beijing Technology and Business University, Beijing 100048, China

**Keywords:** *Fusarium oxysporum*, *Fusarium solani*, fungicides, biocontrol agents, virulence detection

## Abstract

Root rot is one of the most serious diseases affecting *Astragalus membranaceus*, significantly reducing its yield and quality. This study focused on root rot in *Astragalus membranaceus* var. *mongholicus*. Pathogenic fungi were isolated and identified. The pathogenicity of seven strains of pathogenic fungi was verified according to Koch’s postulates. The inhibitory effects of eight classic fungicides and nine strains of biocontrol agents on the pathogenic fungi were determined using the mycelial growth rate method. Through morphological and ITS phylogenetic analyses, strains CDF5, CDF6, and CDF7 were identified as *Fusarium oxysporum*, while strains CDF1, CDF2, CDF3, and CDF4 were identified as *Fusarium solani*. Indoor virulence tests showed that, among the eight tested fungicides, carbendazim exhibited the strongest inhibitory effect on the mycelial growth of both *F. oxysporum* and *F. solani*, with a half-maximal effective concentration (EC_50_) value of (0.44 ± 0.24) mg/mL, making it a highly promising chemical agent for the control of *A. membranaceus* var. *mongholicus* root rot. Among the nine biocontrol agents, KRS006 showed the best inhibitory effect against the seven pathogenic strains, with an inhibition rate ranging from 42.57% to 55.51%, and it can be considered a candidate strain for biological control. This study identified the biocontrol strain KRS006 and the chemical fungicide carbendazim as promising core agents for the biological and chemical control of *A. membranaceus* var. *mongholicus* root rot, respectively, providing a theoretical foundation for establishing a dual biocontrol–chemical control strategy. Based on the excellent performance of the biocontrol bacteria and fungicides in the pathogen control tests, future research should focus on field trials to verify the synergistic effect of this integrated control strategy and clarify the interaction mechanism between the antibacterial metabolites produced by the biocontrol bacteria KRS006 and carbendazim. Additionally, continuous monitoring of the evolution of *Fusarium* spp. resistance to carbendazim is critical to ensure the long-term sustainability of the integrated control system.

## 1. Introduction

*Astragalus membranaceus* (Fisch.) Bge. var. *mongholicus* (Bge.) Hsiao or *Astragalus membranaceus* (Fisch.) Bge. is the dried root of a leguminous plant [1], which is a perennial herb and an important traditional Chinese medicinal material. With the increasing demand for *Astragalus membranaceus*, numerous issues have emerged that affect the development of the *Astragalus* industry, impacting both the yield and quality of the herb. Currently, there is a scarcity of research on the pathogenesis and causes of root rot in *A. membranaceus*. Therefore, it is imperative to conduct in-depth studies on the pathogens of *Astragalus* root rot.

Root rot is a common soil-borne disease. Root rot pathogenic fungi can harm a variety of crops, such as pumpkin [2], pea [3], gastrodia [4], and codonopsis [5]. Due to the high host specificity of *F. oxysporum* [6], it infects roots and stems, causing them to rot. This is particularly devastating for medicinal materials whose roots are the medicinal parts, such as *Astragalus*, severely affecting their yield and quality. These pathogenic fungi include *F. oxysporum* [7,8,9,10,11,12], *F. solani* [7,8,9,10,11,12], *F. acuminatum* [7], and *F. equiseti* [13,14]. *Astragalus* root rot caused by *F. solani* and *F. oxysporum* has been found in major *Astragalus* planting areas in China, such as Gansu, Inner Mongolia, Shaanxi, and Shanxi. After infection by *F. solani* and *F. oxysporum*, the disease develops rapidly and has a wide range of transmission routes. Chen et al. [12] isolated the pathogenic fungi *Rhizoctonia solani* from *Astragalus* suffering from root rot in Weiyuan County. Guan et al. [15] first discovered that the pathogenic fungi of *Astragalus* root rot in Northeast China was *Dactylonectria torresensis*. Qi et al. [16] first reported that *Clonostachys rosea* was the pathogenic fungi of *Astragalus* root rot in Sada Field, Minhe County, Haidong City, Qinghai Province. Among them, the most harmful pathogenic fungi are *F. solani* and *F. oxysporum*.

At present, chemical control is the main method of prevention and treatment. However, chemical control causes significant harm to the environment. The types and amounts of fungicides used by farmers in the planting process were uneven. These fungicides include carbendazim, chlorothalonil, captan, mancozeb, prochloraz, tebuconazole, and metalaxyl-mancozeb [17]. Carbendazim and tebuconazole have been found to have good inhibitory effects on *F. oxysporum* and *F. solani*, which are the main pathogenic fungi of root rot [18]. The application of 1.25 mg/mL thiophanate-methyl had a significant inhibitory effect on *F. oxysporum* and *F. solani*, with the inhibition rates reaching 85.3% and 84.4%, respectively [19]. The only fungicide registered for *Astragalus* is tebuconazole. Therefore, it is necessary to explore fungicides for the prevention and treatment of *Astragalus* root rot. In terms of biological control, *Bacillus* is currently the most widely used. The diameter of the inhibitory region of the *Bacillus* strain SXKF16-1 against *F. solani* and *F. acuminearum* was (25.90 ± 1.18) mm and (25.86 ± 1.85) mm, respectively, which has a long-lasting inhibitory effect on mycelium growth [20]. The growth inhibition rate of the dominant *Bacillus* strain AS1 against *F. oxysporum* was more than 35%, and the results of a pot experiment showed that the control effect of this strain against *Astragalus* root rot could reach 46.86% [21]. In this study, pathogenic fungi from the roots of *Astragalus* infected with root rot collected from *Astragalus* planting bases were isolated and identified. Eight fungicides, including the registered fungicide tebuconazole for *Astragalus*, were selected to determine their toxicity against the pathogenic fungi of *Astragalus* root rot. The fungicides with higher toxicity were preliminarily screened out, providing a reference for the prevention and control of *Astragalus* root rot.

## 2. Materials and Methods

### 2.1. Occurrence of Root Rot in Astragalus membranaceus var. mongholicus

Root rot significantly damages the stem base and roots of *A. membranaceus* var. *mongholicus*. The symptoms of infected plants are primarily characterized by the following (Figure 1): leaves turn yellow and become soft, gradually wilting and progressively falling off. The stem base and roots gradually develop reddish-brown dry rot, with the epidermis becoming easily detached. The main root and stem base exhibit reddish-brown dry rot, with cracked or longitudinally split stripes. In severe cases, most leaves wilt and fall off, and the entire plant can die.

### 2.2. Sample Collection and Screening of Test Agents and Biocontrol Bacteria

The tested strains were collected from diseased plants with typical symptoms in a severe root rot-infected field of *A. membranaceus* var. *mongholicus* in Chengde City, Hebei Province, China, in 2022; a total of 12 diseased samples were collected. The latitude and longitude coordinates were 117°15′19″ N and 41°29′30″ E. The tested *Astragalus* species was *A. membranaceus* (Fisch.) Bge. var. *mongolicus* (Bunge) Hsiao. The culture medias were as follows: potato dextrose agar (PDA), consisting of 200 g of potato, 20 g of dextrose, 18 g of agar, and 1000 mL of distilled water; potato dextrose broth (PDB), consisting of 200 g of potato, 20 g of dextrose, and 1000 mL of distilled water; and Luria–Bertani (LB) medium, consisting of 10 g of tryptone, 5 g of yeast extract, 10 g of NaCl, and 1000 mL of distilled water. The tested chemicals were as follows: 98% difenoconazole, 98% chlorothalonil, 97% bromothalonil, 97% thiophanate-methyl, 96% fludioxonil, 97% hymexazol, 96% tebuconazole, and 97% carbendazim technical grade, which were purchased from Beijing Huike Tongchuang Scientific Instrument Co., Ltd. (Beijing, China), and acetonitrile (mass spectrometry grade), which was purchased from Thermo Fisher Scientific (Waltham, MA, USA).

The tested strains were provided by the research team of Professor Chen Jieyin at the Institute of Plant Protection, Chinese Academy of Agricultural Sciences. The bacterial strains used in this study were KRS002, KRS003, KRS004, KRS006, KRS008, KRS009, KRS013, KRS014, and KRS034. KRS006 was identified as *Serratia premortalis*; KRS004, KRS008, KRS009, and KRS034 were identified as *Bacillus halodurans*; and KRS002, KRS003, KRS013, and KRS014 were identified as *Bacillus pumilus* [22].

Furthermore, 50 and 15 mL plastic centrifuge tubes and 0.22 μm microporous membranes were purchased from Shanghai Anpu Experimental Technology Co., Ltd. (Shanghai, China). A blood cell counting plate was purchased from Beijing Solaibao Technology Co., Ltd. (Beijing, China).

### 2.3. Isolation and Identification of Pathogens

#### 2.3.1. Isolation of Pathogens

The tissue isolation method was adopted and optimized [23]. The rhizomes of the diseased *Astragalus* samples collected from the field were washed clean. The roots were soaked in 75% ethanol for 1 min and then rinsed three times with sterile water. Subsequently, the roots were soaked in 10% sodium hypochlorite for 2 min, rinsed with sterile water to remove the excess disinfectant, and air-dried. During this process, the disinfection time could be appropriately adjusted according to the thickness of the *Astragalus* roots. The disinfected roots were then cut into small segments of 1–2 cm using sterilized scissors and inoculated on PDA plates. The inoculated plates were incubated at 28 °C in the dark. After 2–3 days, the most vigorous mycelium from the edge of uncontaminated colonies on the medium was selected and transferred to new PDA plates for purification to obtain pure cultures of the pathogen, which were then stored at 4 °C.

#### 2.3.2. Determination of the Pathogenicity of Pathogenic Fungi

The purified strain was inoculated on a PDA plate and cultured at 25 °C for 5–7 days. Then, the mycelium at the edge of the colonies was inoculated into PDB medium and cultured at a temperature of 25 °C and 180 r/min for 5–7 days. The culture solution was filtered with four layers of gauze to obtain spore liquid; then, the number of spores in the suspension was calculated using a blood cell counting plate and finally diluted to about 5.0 × 10^6^ spores/mL. Seedlings of *Astragalus* were treated with the spore suspension using the root drenching inoculation method. After disinfection (with 75% alcohol and 10% sodium hypochlorite alternately for 15 min, followed by washing with sterile water 3–5 times), the seeds of *A. membranaceus* var. *mongholicus* were sown in seedling pots filled with sterilized soil, with 10–15 seeds sown in each pot. When the first cotyledon of *A. membranaceus* var. *mongholicus* seedlings grew, 30 mL of the spore suspension with a concentration of 5.0 × 10^6^ spores/mL was poured into each seedling pot, and the negative control was added dropwise. Inoculated plant materials were cultured in a greenhouse at 25 °C, and the incidence was observed and recorded after 10 days.

#### 2.3.3. Morphological Observation of Pathogenic Fungi

The purified pathogen was inoculated into PDA medium and cultured in the dark at 28 °C for 5 days. During the cultivation process, the morphology of the colonies was observed and recorded. When the mycelium covered the entire Petri dish, a sterile punch with a diameter of 5 mm was used to cut out mycelial plugs, which were then inoculated into PDB medium and cultured on a shaker (28 °C, 160 r/min) for 5 to 7 days. Once the strain produced spores, the microscopic morphology of the pathogen was observed under a microscope. Microscopes and a computer were used to observe and measure spores. The microscope slide was photographed using the computer, and the standard length line segment was marked as the basis for measuring the size of the spores.

#### 2.3.4. Molecular Identification of Pathogenic Fungi

The pathogenic fungi were identified according to the “Laboratory Guide for the Identification of Major Species” [24]. The mycelium grown on PDA medium was collected and thoroughly ground with liquid nitrogen. DNA was then extracted using a DNA rapid extraction kit (Sangon Biotech Co., Ltd., Shanghai, China). The fungal ITS region was amplified using the universal primers ITS1/ITS4 (ITS1: TCCGTAGGTCCTGCGG; ITS4: TCCTCCGCTTATTGATATGC) through PCR. The amplification system consisted of the following: 1 μL of template, 25 μL of 2×Taq MasterMix, 1 μL each of ITS1/ITS4, and 22 μL of ddH_2_O, with a total reaction volume of 50 μL. The amplification program was as follows: initial denaturation at 94 °C for 4 min, followed by 32 cycles of denaturation at 94 °C for 1 min, annealing at 54 °C for 30 s, and extension at 72 °C for 1 min, with a final extension at 72 °C for 10 min.

Based on the preliminary *Fusarium* genus identification using ITS sequences, molecular characterization of the pathogen was performed by amplifying the EF-1α gene with the primers EF1/EF2 (EF1: ATGGGTAAGGAGGACAAGAC; EF2: GGAAGTACCAGTGATCATGTT). The amplification system consisted of the following: 2 μL of template, 25 μL of 2×Taq MasterMix, 1 μL each of EF1/EF2, and 21 μL of ddH_2_O, with a total reaction volume of 50 μL. The amplification program was as follows: initial denaturation at 95 °C for 5 min, followed by 35 cycles of denaturation at 95 °C for 1 min, annealing at 58 °C for 45 s, and extension at 72 °C for 2 min, with a final extension at 72 °C for 7 min.

A 0.01% agarose gel was prepared. Subsequently, agarose gel electrophoresis was performed with a voltage setting of 120 V and a duration of 35 min to detect the presence of the target band. The PCR stock solution with the target band detection meeting expectations was entrusted to Beijing Qingke Biotechnology Co., Ltd. (Beijing, China), for sequencing. The sequencing results obtained were used to construct a phylogenetic tree using the neighbor-joining method.

### 2.4. Determination of the Sensitivity of Pathogenic Fungi to Fungicides

Acetonitrile (concentration ≥ 99.9%) was used to prepare the eight tested fungicides in the mother solutions with the mass concentrations shown in Table 1, which were then diluted into different series of mass concentrations of solutions for testing. The mycelial growth rate method was employed to determine the inhibitory effects of the eight fungicides on the mycelial growth of the seven tested fungal strains that were isolated. The tested fungal strains were inoculated on PDA plates and pre-cultured at 25 °C in the dark for 5 days. Mycelial discs with a diameter of 5 mm were taken and inoculated on PDA plates with a diameter of 90 mm and containing a series of mass concentrations of fungicides. Plates without fungicides were used as controls, and each treatment was repeated three times. The plates were incubated at 25 °C in the dark for 8 days. The colony diameter (D) of each treatment was measured using the cross method, and the inhibition rate of each fungicide at different concentrations against the mycelial growth of the pathogen was calculated. The inhibition rate was taken as the vertical coordinates (Y), and the logarithm of the mass concentration of the fungicide was taken as the horizontal coordinates (X). The toxicity regression equation Y = aX + b, the correlation coefficient (r), and the effective inhibitory median concentration (EC_50_, μg/mL) of the fungicide were determined. The average EC_50_ values of CDF1 to CDF7 for each fungicide were calculated (average EC_50_, μg/mL).

### 2.5. Determination of the Sensitivity of Pathogenic Fungi to Biocontrol Bacteria

#### 2.5.1. Activation of Pathogenic Fungi and Probiotic Bacteria

The test strains were inoculated on PDA plates and cultured at 25 °C in the dark for 5 days before use. The preserved test biocontrol bacteria were inoculated into LB medium and cultured at 28 °C with a speed of 180 r/min for 1 to 3 days. The bacterial solution was observed under a microscope to ensure that there was no contamination. During this process, the OD600 value of each culture medium was measured when it reached 1.

#### 2.5.2. Evaluation of the Efficacy of Biocontrol Bacteria Against Pathogenic Fungi

The inhibitory effect of 9 biocontrol bacteria on the mycelial growth of the 7 test strains CDF1 to CDF7 was determined using the mycelial growth rate method. First, 5 mm diameter test pathogen discs were inoculated on 90 mm diameter PDA plates. The biocontrol bacteria were streaked in a parallel diameter direction 20 mm away from the center of the plate. A control plate without inoculation of the biocontrol bacteria was used as a reference. Each treatment was repeated 3 times. The plates were placed in a dark environment at 25 °C for 5 to 7 days. The radial distance between the center of the colony plate and the outer edge of the colony was measured. The distance of the biocontrol treatment and the control was recorded as “a” and “b”, respectively. The formula for calculating the related inhibition rate (IR) is IR (%) = (b − a)/b × 100% (6 − 2), where IR represents the percentage of colony growth inhibition; b represents the diameter of the control colony, in mm; and a represents the diameter of the treated colony, in mm.

### 2.6. Data Analysis

The colony diameter (D) of each treatment was measured using the cross-multiplication method, and the inhibition rates of different concentrations of each fungicide on the mycelial growth of the pathogen were calculated. The probability values of the inhibition rates were taken as the vertical coordinates (Y), and the logarithmic values of the drug concentration were taken as the horizontal coordinates (X). The toxicity regression equation Y = aX + b, the correlation coefficient (R), and the effective inhibitory concentration (EC_50_, mg/L) of the drug were obtained. The test data were the average values of three repeated treatments. The original data were summarized and statistically analyzed in Excel; a variance analysis was performed using SPSS17.0; Pearson and Spearman correlation analyses were conducted using Origin2019; and charts and their combinations were drawn and assembled using Origin2019 and Ai2020.

## 3. Results

### 3.1. Morphological Observation and Pathogenicity Determination of Pathogenic Fungi

The tissue isolation method was employed for the isolation and purification of pathogens from the diseased roots of *A. membranaceus* var. *mongholicus*, yielding a total of seven strains. The colony morphology of the pathogen and the morphology of the conidia are shown in Figure 2 and Figure 3, respectively. The morphologies of large conidia and chlamydospores are depicted in Figure 4. The strains CDF1, CDF2, CDF3, and CDF4 were identified as *Fusarium solani*.

The colony of CDF1 was circular, with white, cottony mycelium. The aerial mycelium was thin and velvety, ranging from white to light gray. The colony appeared white, with a rough and dry surface. Brownish-red pigments could be observed in the center of the colony’s reverse side. The small conidia were oval, kidney-shaped, etc., with a relatively thick wall and 0 to 1 septa.

The colony of CDF2 was nearly circular with an irregular edge. The central mycelium was yellowish-brown, while the edge was white, with the color deepening toward the center. The reverse side of the colony was dark brown. The small conidia were oval, short, and sausage-shaped, with most being aseptate and a few having one septum. They were transparent and smooth.

The colony of CDF3 was circular, with white or light gray mycelium. The center was compact, while the edge was sparse. The reverse side of the colony was also white or gray. The large conidia were robust and slightly curved into a falcate shape, with 2 to 4 transverse septa. The small conidia were mostly oval or kidney-shaped.

The colony of CDF4 was nearly circular with an irregular edge. The mycelium was dense, and the surface was rough. Yellowish-brown pigments could be observed from the reverse side of the colony. The large conidia were slightly curved into a falcate shape, with 2 to 3 transverse septa. The small conidia were oval or kidney-shaped.

CDF5, CDF6, and CDF7 were identified as *Fusarium oxysporum*. The colony of CDF5 initially appeared white and then gradually turned pink with a slight purple hue. Yellowish-brown pigments could be observed on the reverse side of the colony. The small conidia were cylindrical or kidney-shaped.

The colony of CDF6 was circular, with a prickly irregular edge. The mycelium ranged from light pink to purple, with a dense center and sparse edges. The reverse side of the colony was also pink to light purple. The large conidia were robust and slightly curved into a falcate shape, with 3 to 5 transverse septa. The small conidia were mostly oval or kidney-shaped.

The colony of CDF7 initially appeared white and then gradually turned purple, producing pigments that ranged from purple to dark purple. A large number of small conidia, which were oval or club-shaped, were produced on the aerial mycelium. The large conidia were falcate, mostly with 0 to 1 septa.

Verification through Koch’s postulates confirmed that the seven isolated pathogenic strains were pathogenic to *A. membranaceus* var. *mongholicus*.

### 3.2. Phylogenetic Analysis

Seven strains of pathogenic fungi were identified based on morphology and molecular biology. Following an ITS-based identification of the fungal isolate as *Fusarium* spp., the obtained TEF sequences were used to construct a phylogenetic tree. All reference sequences were retrieved from the NCBI GenBank database. Through a phylogenetic analysis, it was found that CDF1, CDF2, CDF3, and CDF4 were *Fusarium solani*, while CDF5, CDF6, and CDF7 were *Fusarium oxysporum* (Figure 5).

### 3.3. Toxicity of Fungicides to F. solani and F. oxysporum

The results of this study show that, among the eight tested fungicides (Table 2), carbendazim exhibited the strongest toxicity against the seven pathogenic strains, with an average EC_50_ value of (0.44 ± 0.24) μg/mL (Figure 6). This was followed by tebuconazole, chlorothalonil, propiconazole, and thiophanate-methyl, with average EC_50_ values ranging from 1.74 to 36.82 μg/mL. The toxicity of bromothalonil was moderate, while that of hymexazol and fludioxonil was relatively low (Table 2).

### 3.4. Control Effect of Biocontrol Bacteria on F. solani and F. oxysporum

The results of this study indicate that, among the nine tested biocontrol strains, KRS006 exhibited the strongest inhibitory effect against strains CDF1 to CDF7 (Figure 7), with an average inhibition rate of 47.73%. This was followed by KRS002, KRS034, KRS014, KRS008, KRS009, KRS004, KRS013, and KRS034, with inhibition rates ranging from 21.17% to 41.97%, 14.77% to 31.73%, 5.80% to 26.58%, 6.79% to 23.9%, 2.83% to 23.92%, 4.48% to 21.78%, 5.84% to 24.49%, and 2.63% to 12.71%, respectively (Table 3). Comparing the inhibition rates against individual pathogenic strains revealed that KRS006 had the highest inhibition rate against all seven pathogenic strains, followed by KRS002. This suggests that KRS006 can be considered a candidate strain for the biological control of root rot in *Astragalus membranaceus* var. *mongholicus*.

## 4. Discussion

In this study, carbendazim demonstrated significantly higher efficacy against *Fusarium oxysporum* and *Fusarium solani* than other fungicides. Although carbendazim has not yet been registered for the control of *Astragalus* root rot, its superior performance makes it a critical candidate for further field trials of fungicide screening. However, the inhibitory effect of thiophanate-methyl on the pathogen of *Astragalus membranaceus* var. *mongholicus* was relatively poor, with an average EC_50_ of 36.82 mg·L^−1^. This result is different from that in He’s study (thiophanate-methyl root-dip treatment of *A. membranaceus* var. *mongholicus* seedlings showed good inhibitory effects on *F. oxysporum* and *F. solani*) [25]. The difference in efficacy may be related to the type of pathogen, application method, and individual differences in the strains.

In the biological control of *Astragalus* root rot, relevant studies have shown that the control effect of the agent *Bacillus subtilis* 04 on the root rot of *Astragalus* is 68.01~80.44% [26]. *Streptomyces cellulolyticus* has exhibited significant inhibitory effects against various *Fusarium* species, with an average antifungal activity exceeding 50% [27]. The plate inhibition rate of *Trichoderma harzianum* against *Fusarium spp.* causing root rot in *Astragalus membranaceus* in Ningxia saline–alkali regions can reach approximately 70% [28]. The nine biocontrol strains screened in this study demonstrated notable control potential. KRS006 exhibited the strongest inhibitory effect against strains CDF1 to CDF7, with an average inhibition rate of 47.73%. This variation may be attributed to significant functional differences among biocontrol strains in plant growth promotion and disease suppression. Currently, an increasing number of studies are employing the combined use of chemical pesticides and biocontrol agents for the control of crop pests and diseases [29]. In this context, the application of a combination of carbendazim and *Bacillus amyloliquefaciens* for the control of branch blight in Koelreuteria paniculata achieved significantly better results than the application of single-agent treatments [30]. The combination of metabolites from *Bacillus amyloliquefaciens* and carbendazim provided good control of peanut brown spot disease [31]. A mixture of yellow–blue fungi fermentation liquid and mancozeb provided effective control of wheat scab [32]. Agricultural Jiaosu (AJ), which is rich in beneficial microorganisms (Bacillus, Pseudomonas, and Lactobacillus) and other substances, showed good inhibitory effects against *F. oxysporum* [33]. Further research could be conducted on the effects of combinations of fungicides and biocontrol bacteria on the control of root rot in *Astragalus membranaceus* var. *mongholicus*.

The pathogens causing root rot in *Astragalus membranaceus* var. *mongholicus* from different origins are complex and exhibit significant biological differences. Currently, research on the pathogenic mechanisms of *Astragalus membranaceus* var. *mongholicus* root rot both domestically and internationally needs to be further strengthened. The long-term and large-scale use of chemical pesticides not only easily leads to excessive pesticide residues and environmental pollution, thereby disrupting ecological balance [34], but also makes pathogens prone to developing drug resistance [35,36]. *Bacillus* spp. are pivotal biocontrol agents of *Astragalus* root rot, and they can produce an array of metabolites that stimulate plant growth and mitigate pathogen attacks through either the direct inhibition of fungal growth or the induction of the plant immune system against pathogens [37]. The biocontrol of *Astragalus* root rot provides eco-friendly and durable management, thereby supporting sustainable herb production. Therefore, the prevention and control strategies for diseases in *Astragalus membranaceus* var. *mongholicus* are moving toward biological control and integrated management. Future research will place a greater emphasis on environmental protection and sustainability to ensure the long-term healthy development of the *Astragalus membranaceus* var. *mongholicus* industry. Through these measures, we can not only protect and enhance the medicinal value of *Astragalus membranaceus* var. *mongholicus* but also ensure its important role in food safety and ecological protection.

## 5. Conclusions

In this study, through pathogen isolation, pathogenicity determination, morphological observation, and an ITS gene sequence analysis, the *Astragalus* root rot pathogens CDF5, CDF6, and CDF7 were identified as *Fusarium oxysporum*, while CDF1, CDF2, CDF3, and CDF4 were identified as *Fusarium solani*. The in vitro control efficacy of fungicides and biocontrol agents against *Fusarium oxysporum* and *Fusarium solani*, the pathogenic fungi that cause root rot in *Astragalus membranaceus*, was evaluated, resulting in the screening of highly effective fungicides and biocontrol strains. The sensitivity of the isolated pathogens to fungicides was tested using the mycelial growth rate method. The results show that carbendazim and tebuconazole exhibited the best inhibitory effects, particularly carbendazim, which demonstrated significantly higher efficacy against *F. oxysporum* and *F. solani* than other fungicides. Therefore, carbendazim can be prioritized for field screening in *Astragalus* root rot control. Among the nine biocontrol strains, KRS006 and KRS002 displayed superior antimicrobial activity, indicating their potential for disease control. The results could provide a foundation for future research on microbial biocontrol formulations. The pathogens that cause *Astragalus* root rot vary significantly in origin and biological characteristics across production regions, and further research is needed to elucidate their pathogenic mechanisms. The long-term and extensive use of chemical fungicides may lead to pesticide residues, environmental pollution, and the development of pathogen resistance. Therefore, the control strategy for *Astragalus* diseases is shifting toward biological control and integrated management to ensure the sustainable development of the *Astragalus* industry.

## Figures and Tables

**Figure 1 jof-11-00544-f001:**
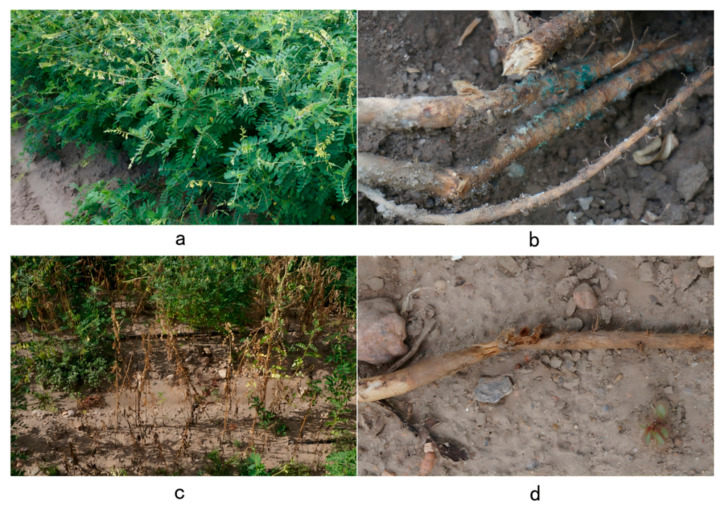
Symptoms of *A. membranaceus* var. *mongholicus* root rot. (**a**) Healthy *A. membranaceus* var. *mongholicus*. (**b**) Onset of *A. membranaceus* var. *mongholicus* root rot. (**c**) Root symptoms in the early and middle stages of the onset of *A. membranaceus* var. *mongholicus* root rot. (**d**) Symptoms of root disease in the late stage of *A. membranaceus* var. *mongholicus* root rot. These photos were taken on 14 August 2022 in Longhua County, Chengde City, Hebei Province.

**Figure 2 jof-11-00544-f002:**
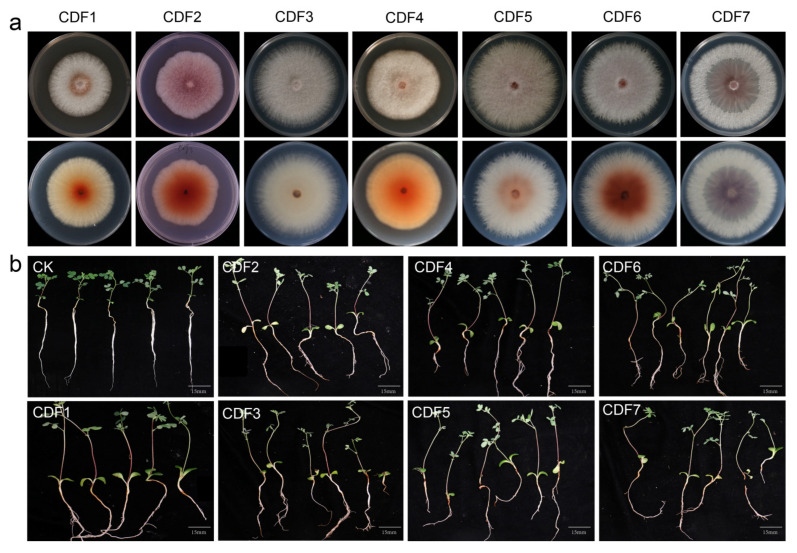
The symptoms of colony morphology and pathogenicity of root rot of *A. membranaceus* var. *mongholicus*. (**a**) The top image shows the positive form of the colony on the potato dextrose agar, and the bottom image shows the back form of the potato dextrose agar. (**b**) The determination result of the pathogenicity of pathogenic fungi.

**Figure 3 jof-11-00544-f003:**
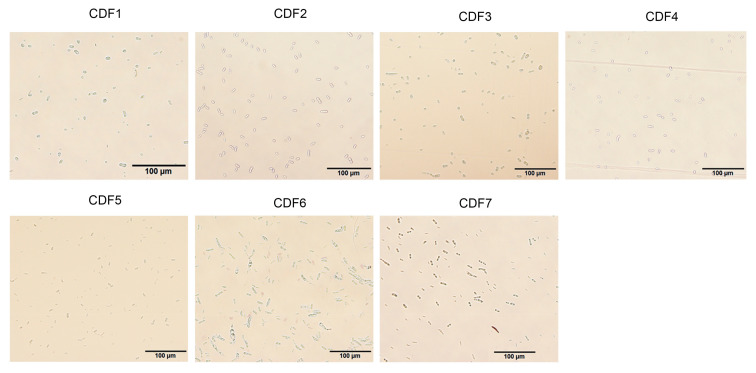
Morphology of the conidia of root rot pathogen of *A. membranaceus* var. *mongholicus*.

**Figure 4 jof-11-00544-f004:**
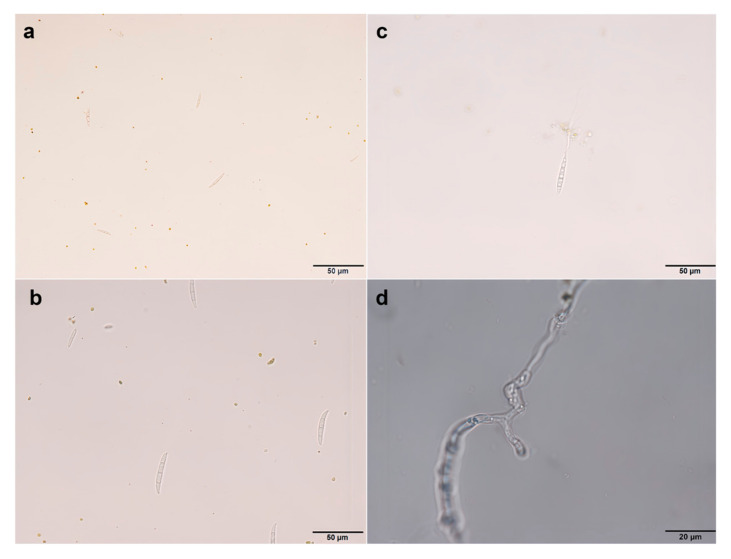
Morphology of macroconidia and chlamydomonas of the root rot pathogen of *Astragalus membranaceus* var. *mongholicus*. (**a**) Morphology of macroconidia of *F. oxysporum*. (**b**) Morphology of macroconidia of *F. solani*. (**c**) Morphology of chlamydospore of *F. oxysporum*. (**d**) Morphology of chlamydospore of *F. solani*.

**Figure 5 jof-11-00544-f005:**
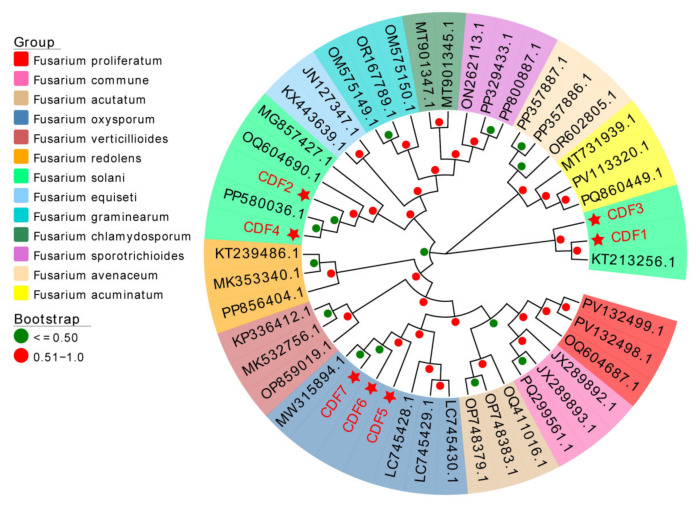
Construction of the phylogenetic tree of *Fusarium* spp. based on EF-1α genes. Notes: The red star in the figure represents the pathogenic fungi isolated in Section 3.1 of this article.

**Figure 6 jof-11-00544-f006:**
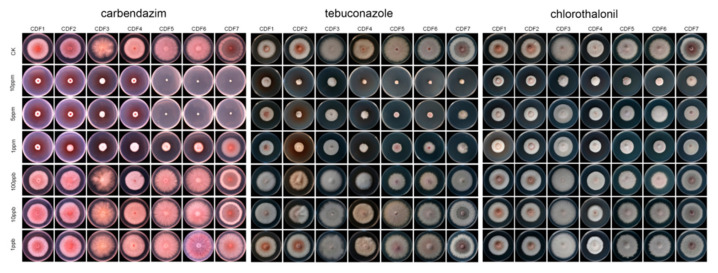
The inhibitory effect of the tested fungicide on pathogenic fungi.

**Figure 7 jof-11-00544-f007:**
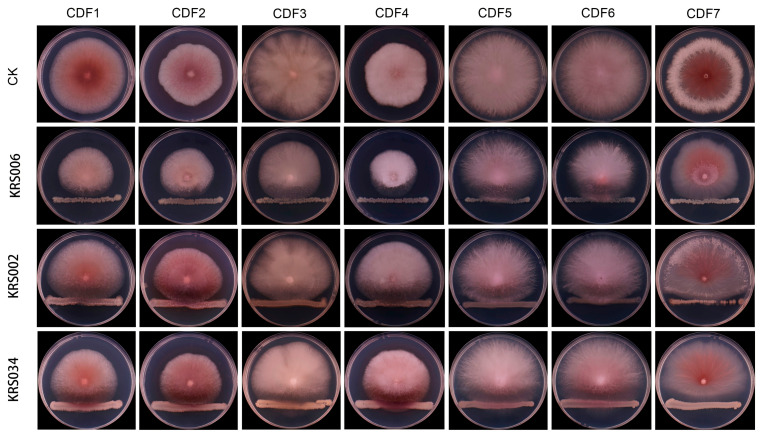
The inhibitory effect of some tested biocontrol bacteria on pathogenic fungi.

**Table 1 jof-11-00544-t001:** Characterization of tested fungicides: active ingredients, chemical formula, classification, and experimental concentrations.

Fungicide (Common Name)	Chemical Formula	Fungicide Class	Initial Concentration (mg/L)	Series Dilution Concentration (mg/L)
Difenoconazole	C_19_H_17_C_l2_N_3_O_3_	Triazol	1000	0.001, 0.01, 0.1, 1, 5, 10
Chlorothalonil	C_8_Cl_4_N_2_	Substitutive benzene	1000	0.001, 0.01, 0.1, 1, 5, 10
Bromothalonil	C_6_H_6_Br_2_N_2_	Bromocyanoalkanes	1000	0.001, 0.01, 0.1, 1, 5, 10
Thiophanate-methyl	C_12_H_14_N_4_O_4_S_2_	Substitutive benzene	1000	0.001, 0.01, 0.1, 1, 5, 10
Hymexazol	C_4_H_5_NO_2_	Triadimefon	1000	0.001, 0.01, 0.1, 1, 5, 10
Tebuconazole	C_16_H_22_ClN_3_O	Triazol	1000	0.001, 0.01, 0.1, 1, 5, 10
Fludioxonil	C_12_H_6_F_2_N_2_O_2_	Phenylpyrrole	1000	0.001, 0.01, 0.1, 0.5, 1, 2
Carbendazim	C_9_H_9_N_3_O_2_	Benzimidazole	1000	0.001, 0.01, 0.1, 1, 5, 10

**Table 2 jof-11-00544-t002:** Virulence regression equations, EC_50_ values, and related parameters of different fungicides against target isolates using a 3-point assay.

Fungicide	Isolate	Virulent Regression Equation	EC_50_/ (μg/mL)	Correlation Coefficient, r	Average EC_50_ Value/(μg/mL)
tebuconazole	CDF1	Y = 0.5368 ∗ X + 4.505	8.36	0.98	1.74
	CDF2	Y = 0.6164 ∗ X + 5.336	0.29	0.96	
	CDF3	Y = 0.6492 ∗ X + 4.855	1.67	0.99	
	CDF4	Y = 0.6911 ∗ X + 5.176	0.56	0.98	
	CDF5	Y = 0.7754 ∗ X + 5.486	0.24	0.98	
	CDF6	Y = 0.8742 ∗ X + 5.427	0.32	0.99	
	CDF7	Y = 0.7390 ∗ X + 5.099	0.73	0.89	
difenoconazole	CDF1	Y = 0.5459 ∗ X + 4.343	15.98	0.99	11.55
	CDF2	Y = 0.4562 ∗ X + 4.481	13.73	0.99	
	CDF3	Y = 0.3752 ∗ X + 4.761	4.34	0.96	
	CDF4	Y = 0.2438 ∗ X + 4.595	45.84	0.99	
	CDF5	Y = 0.5516 ∗ X + 5.362	0.22	0.99	
	CDF6	Y = 0.5149 ∗ X + 5.228	0.38	0.97	
	CDF7	Y = 0.6981 ∗ X + 5.288	0.39	0.99	
chlorothalonil	CDF1	Y = 0.5787 ∗ X + 4.569	5.56	0.98	8.58
	CDF2	Y = 0.4170 ∗ X + 4.414	25.43	0.97	
	CDF3	Y = 0.4801 ∗ X + 4.522	9.90	0.96	
	CDF4	Y = 0.4869 ∗ X + 4.515	9.91	0.98	
	CDF5	Y = 0.5135 ∗ X + 4.834	2.11	0.98	
	CDF6	Y = 0.4439 ∗ X + 4.844	2.25	0.95	
	CDF7	Y = 0.3768 ∗ X + 4.739	4.93	0.96	
bromothalonil	CDF1	Y = 0.3942 ∗ X + 3.960	434.77	0.97	368.73
	CDF2	Y = 0.4215 ∗ X + 3.696	1240.83	0.98	
	CDF3	Y = 0.4000 ∗ X + 3.835	817.52	0.96	
	CDF4	Y = 0.5585 ∗ X + 4.067	46.83	0.93	
	CDF5	Y = 0.4883 ∗ X + 4.478	12.31	0.98	
	CDF6	Y = 0.4701 ∗ X + 4.381	20.74	0.96	
	CDF7	Y = 0.3285 ∗ X + 4.701	8.13	0.96	
hymexazol	CDF1	Y = 0.2996 ∗ X + 4.116	892.48	0.99	959.33
	CDF2	Y = 0.3008 ∗ X + 4.187	504.42	0.98	
	CDF3	Y = 0.3277 ∗ X + 4.076	660.16	0.98	
	CDF4	Y = 0.3298 ∗ X + 3.925	1817.82	0.97	
	CDF5	Y = 0.3508 ∗ X + 3.917	1222.44	0.98	
	CDF6	Y = 0.3316 ∗ X + 3.972	1259.28	0.97	
	CDF7	Y = 0.4431 ∗ X + 3.868	358.70	0.99	
thiophanate- methyl	CDF1	Y = 0.3701 ∗ X + 4.473	26.54	0.94	36.82
	CDF2	Y = 0.4092 ∗ X + 4.504	16.30	0.98	
	CDF3	Y = 0.4116 ∗ X + 4.587	10.08	0.96	
	CDF4	Y = 0.3127 ∗ X + 4.539	29.80	0.99	
	CDF5	Y = 0.5373 ∗ X + 4.457	10.25	0.99	
	CDF6	Y = 0.3421 ∗ X + 4.521	25.13	0.94	
	CDF7	Y = 0.2606 ∗ X + 4.441	139.65	0.97	
fludioxonil	CDF1	Y = 0.3546 ∗ X + 3.716	4178.12	0.90	2872.67
	CDF2	Y = 0.2119 ∗ X + 4.458	361.25	0.88	
	CDF3	Y = 0.3290 ∗ X + 3.847	3195.65	0.91	
	CDF4	Y = 0.2238 ∗ X + 4.210	3387.95	0.85	
	CDF5	Y = 0.3708 ∗ X + 3.646	4482.95	0.91	
	CDF6	Y = 0.3646 ∗ X + 3.702	3631.33	0.91	
	CDF7	Y = 0.3765 ∗ X + 3.893	871.44	0.91	
carbendazim	CDF1	Y = 0.9079 ∗ X + 5.303	0.46	0.96	0.44
	CDF2	Y = 0.7022 ∗ X + 5.254	0.43	0.98	
	CDF3	Y = 0.8832 ∗ X + 5.053	0.87	0.96	
	CDF4	Y = 0.6784 ∗ X + 5.196	0.51	0.97	
	CDF5	Y = 1.336 ∗ X + 5.778	0.26	0.93	
	CDF6	Y = 1.356 ∗ X + 5.775	0.27	0.92	
	CDF7	Y = 1.177 ∗ X + 5.658	0.28	0.85	

Notes: The average EC_50_ value/(μg/mL) in the table represents the average EC_50_ value of one fungicidal agent against seven pathogenic fungi.

**Table 3 jof-11-00544-t003:** The inhibition rate of biocontrol bacteria on pathogenic fungi.

Treatment	Inhibition Rate of Fungal Colony Growth (%)						
	CDF1	CDF2	CDF3	CDF4	CDF5	CDF6	CDF7
KRS002	37.70 ± 2.18	33.06 ± 1.31	40.49 ± 2.22	21.17 ± 3.23	38.59 ± 1.24	29.01 ± 3.03	41.91 ± 1.43
KRS003	12.12 ± 1.82	1.74 ± 0.99	10.67 ± 0.80	6.78 ± 0.93	4.55 ± 0.75	4.03 ± 0.40	2.63 ± 0.34
KRS004	17.59 ± 0.47	21.78 ± 0.99	8.01 ± 0.65	10.52 ± 0.94	9.14 ± 1.13	4.48 ± 0.69	10.91 ± 1.21
KRS006	55.51 ± 3.34	44.83 ± 0.27	49.42 ± 0.25	47.60 ± 0.44	42.57 ± 0.49	45.70 ± 0.74	46.39 ± 1.30
KRS008	23.79 ± 2.62	16.96 ± 2.95	10.29 ± 3.33	13.58 ± 4.78	6.79 ± 2.24	7.25 ± 0.552	11.18 ± 0.59
KRS009	20.46 ± 0.50	21.47 ± 0.68	23.92 ± 0.30	7.28 ± 0.47	5.17 ± 0.19	2.83 ± 0.18	4.76 ± 0.20
KRS013	24.49 ± 3.45	10.43 ± 0.51	15.21 ± 1.29	9.68 ± 0.40	5.89 ± 1.43	5.84 ± 0.28	10.20 ± 1.05
KRS014	26.58 ± 3.06	17.75 ± 2.56	17.11 ± 2.57	8.50 ± 2.66	14.09 ± 4.23	19.03 ± 5.16	5.80 ± 2.11
KRS034	24.80 ± 1.67	21.88 ± 3.79	19.33 ± 1.09	19.02 ± 4.53	14.77 ± 3.28	15.44 ± 3.25	31.73 ± 5.31

## Data Availability

The original contributions presented in this study are included in the article. Further inquiries can be directed to the corresponding author.

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
