# Peer review of "Identifying Key Pathogens and Effective Control Agents for Astragalus membranaceus var. mongholicus Root Rot"

_jof, 2025, doi:10.3390/jof11070544_

Round 1

Reviewer 1 Report (Previous Reviewer 1)

After analysing the article. The authors met all my requests

After analysing the article. The authors met all my requests:

This study focused on the root rot of Astragalus membranaceus var. mongholicus, Pathogenic fungi were isolated and identified.

The article has improved, but still lacks a few observations.

In the abstract I suggest you put an introductory sentence before the aim of the article.

All abbreviations in the abstract must be explained before abbreviating. The reader won't know what they mean. The abstract should be revised. Even the scientific names.

In pictures with photos, as in figure 1,2 you should put a size bar.

The titles of tables in particular, such as table 2, should be improved and be self-explanatory.

Author Response

Comments 1: This study focused on the root rot of Astragalus membranaceus var. mongholicus, Pathogenic fungi were isolated and identified. The article has improved, but still lacks a few observations.

Response 1: Thanks for your acknowledgement for our study. Your comments were carefully addressed to improve the methodology rigorousness and writing readability. 

Comments 2:In the abstract l suggest you put an introductory sentence before the aim of the article.

Response 2: Thanks for your acknowledgement for our study. Your comments were carefully addressed to improve the methodology rigorousness and writing readability. We have added an introductory sentence before the aim of the article in the abstract as per the suggestion. The specific changes are as shown below:

Root rot is one of the most serious diseases affecting Astragalus membranaceus, significantly reducing its yield and quality. (Page 1, Lines 14-15)

Comments 3:All abbreviations in the abstract must be explained before abbreviating. The reader won't know what they mean. The abstract should be revised. Even the scientific names.

Response 3: Thanks for your acknowledgement for our study. Your comments were carefully addressed to improve the methodology rigorousness and writing readability. We have checked the abbreviations in the article and provided the full names before their appearance. The specific changes are as shown below:

a half-maximal effective concentration (EC50) (Page 1, Lines 22)

Comments 4:In pictures with photos, as in figure 1,2 you should put a size bar.

Response 4: Thanks for your acknowledgement for our study. Your comments were carefully addressed to improve the methodology rigorousness and writing readability. Figure 2 is provided to supplement the experimental data in the text and further explain the results. The purpose of Figure 1 is to demonstrate the field incidence of Astragalus root rot through an on-site photograph, not to display the size traits of Astragalus plants.

Comments 5:The titles of tables in particular, such as table 2, should be improved and be self-explanatory.

Response 5: Thanks for your acknowledgement for our study. Your comments were carefully addressed to improve the methodology rigorousness and writing readability. We have made improvements to the titles of the tables in the article as per the suggestions, making them self-explanatory. The specific changes are as shown below:

  • Table 1. Characterization of Tested Fungicides: Active Ingredients, Chemical Formula, Classification, and Experimental Concentrations (Pages 6, Lines 167)
  • Table 2. Irulence Regression Equations, EC50Values, and Related Parameters of Different Fungicides against Target Isolates. (Pages 10, Lines 252)

Reviewer 2 Report (Previous Reviewer 3)

The manuscript is revised in a modified version of a previously reviewed work on two occasions, JOF-3666894. This new version addresses the previously made observations. 

This updated manuscript improved the overall clarity and depth of the research. The updated manuscript incorporates feedback from the review process, presenting the findings in a more comprehensive manner.

Author Response

Comments 1: This updated manuscript improved the overall clarity and depth of the research. The updated manuscript incorporates feedback from the review process, presenting the findings in a more comprehensive manner.

Response 1: Thanks for your acknowledgement for our study.

This manuscript is a resubmission of an earlier submission. The following is a list of the peer review reports and author responses from that submission.

Round 1

Reviewer 1 Report

Identifying Key Pathogens and Effective Control Agents for Astragalus membranaceus var. mongholicus Root Rot

This study focused on the root rot of Astragalus membranaceus var. mongholicus, Pathogenic fungi were isolated and identified.

The article has improved, but still lacks a few observations.

In the abstract I suggest you put an introductory sentence before the aim of the article.

All abbreviations in the abstract must be explained before abbreviating. The reader won't know what they mean. The abstract should be revised. Even the scientific names.

In pictures with photos, as in figure 1,2 you should put a size bar.

The titles of tables in particular, such as table 2, should be improved and be self-explanatory.

Identifying Key Pathogens and Effective Control Agents for Astragalus membranaceus var. mongholicus Root Rot

This study focused on the root rot of Astragalus membranaceus var. mongholicus, Pathogenic fungi were isolated and identified.

The article has improved, but still lacks a few observations.

In the abstract I suggest you put an introductory sentence before the aim of the article.

All abbreviations in the abstract must be explained before abbreviating. The reader won't know what they mean. The abstract should be revised. Even the scientific names.

In pictures with photos, as in figure 1,2 you should put a size bar.

The titles of tables in particular, such as table 2, should be improved and be self-explanatory.

Author Response

Comments 1: Identifying Key Pathogens and Effective Control Agents for Astragalus membranaceus var. mongholicus Root Rot

Response: Thanks for your acknowledgement for our study. Your comments were carefully addressed to improve the methodology rigorousness and writing readability. 

Comments 2:This study focused on the root rot of Astragalus membranaceus var. mongholicus, Pathogenic fungi were isolated and identified. 

Response: Thanks for your acknowledgement for our study. Your comments were carefully addressed to improve the methodology rigorousness and writing readability. 

Comments 3:The article has improved, but still lacks a few observations. 

Response: Thanks for your acknowledgement for our study. Your comments were carefully addressed to improve the methodology rigorousness and writing readability. 

Comments 4:In the abstract I suggest you put an introductory sentence before the aim of the article. 

Response: Thanks for your acknowledgement for our study. We have added an introductory sentence before the aim of the article in the abstract. As follows: Based on the excellent performance of biocontrol bacteria and fungicides in the pathogen control tests, future research should focus on field trials to verify the synergistic effect of this integrated control strategy and clarify the interaction mechanism be-tween the antibacterial metabolites produced by KRS006 and carbendazim. (Page 1, Lines 31-35)

Comments 5:All abbreviations in the abstract must be explained before abbreviating. The reader won't know what they mean. The abstract should be revised. Even the scientific names. 

Response: Thanks for your acknowledgement for our study. We have added full names for the abbreviations in the abstract before each abbreviation. As following: Through morphological and ITS phylogenetic analysis, strains CDF5, CDF6, and CDF7 were identified as Fusarium oxysporum, while strains CDF1, CDF2, CDF3, and CDF4 were identified as Fusarium solani. (Page 1, Lines 19-21)

Comments 6:In pictures with photos, as in figure 1,2 you should put a size bar. 

The titles of tables in particular, such as table 2, should be improved and be self-explanatory.

Response: Thanks for your acknowledgement for our study. We have added a size column in Figure 2 and improved the title of Table 2. The title of table 2: Inhibition of Fusarium strains CDF1 on CDF7 mycelial growth by eight fungicides (Page 8, Lines 276-277; Page 10, Lines 307)

Reviewer 2 Report

Materials and Methods.
Please, add all identification guides for the fungal taxonomy which were used for identification. In which systems names and systematic positions of fungi are given?  MycoID, Westerdijk Fungal Biodiversity Institute and The National Center for Biotechnology Information (GenBank )?

Please, add a Table with NCBI number strains used to construct the phylogenetic tree.
Results:
The images of Figures 3 and 4 need to be redone. It still very poor.

Discussion: There was also no discussion on some important topics. A comparative characteristic with other biocontrol agents should be added. 

In conclusion, is it worth to describe in detail the advantage of this test in comparison with the other available? Will this method be a universal?

The authors need to check the entire manuscript for grammar and spelling.

Materials and Methods.
Please, add all identification guides for the fungal taxonomy which were used for identification. In which systems names and systematic positions of fungi are given?  MycoID, Westerdijk Fungal Biodiversity Institute and The National Center for Biotechnology Information (GenBank )?

Please, add a Table with NCBI number strains used to construct the phylogenetic tree.
Results:
The images of Figures 3 and 4 need to be redone. It still very poor.

Discussion: There was also no discussion on some important topics. A comparative characteristic with other biocontrol agents should be added. 

In conclusion, is it worth to describe in detail the advantage of this test in comparison with the other available? Will this method be a universal?

The authors need to check the entire manuscript for grammar and spelling.

Author Response

Comments 1: Please, add all identification guides for the fungal taxonomy which were used for identification. In which systems names and systematic positions of fungi are given?  MycoID, Westerdijk Fungal Biodiversity Institute and The National Center for Biotechnology Information (GenBank )?

Response: Thanks for your acknowledgement for our study. We had added all identification guides for the fungal taxonomy which were used for identification. As following: The pathogenic bacteria were identified according to "Laboratory Guide for the Identification of Major Species"[23]. (Page 4, Lines 162-163)

Reference:

[23] Gams, W. Laboratory guide to the identification of the major species. Netherlands Journal of Plant Pathology 84, 84 (1978). https://doi.org/10.1007/BF01976412

Comments 2: Please, add a Table with NCBI number strains used to construct the phylogenetic tree.

Response: Thanks for your acknowledgement for our study. In Figure 5, the serial numbers of the strains used to construct the phylogenetic tree are already provided in detail. Adding a table would result in duplication of content with the information in the figure 5. 

Comments 3: The images of Figures 3 and 4 need to be redone. It still very poor.

Response: Thanks for your acknowledgement for our study. Your comments were carefully addressed to improve the methodology rigorousness and writing readability. We have made every effort to redo the pictures, but due to the equipment limitations, the clarity of the pictures can only reach the current level. We are also striving to find better equipment to improve the current result.

Comments 4: Discussion: There was also no discussion on some important topics. A comparative characteristic with other biocontrol agents should be added. 

Response: Thanks for your acknowledgement for our study. Your comments were carefully addressed to improve the methodology rigorousness and writing readability. We have added relevant content in discussion. In the biological control of Astragalus root rot, relevant studies have shown that the control effect of Bacillus subtilis 04 agent on the root rot of astragalus was 68.01%~80.44% [26]. Streptomyces cellulolyticus exhibits significant inhibitory effects against various Fusarium species, with an average antifungal activity exceeding 50%[27]. The plate inhibition rate of Trichoderma harzianum against Fusarium spp. causing root rot in Astragalus membranaceus in Ningxia saline-alkali regions can reach approximately 70% [28].

[26] Fan, L.S., Huang, Z.B., Jin, X.J., et al. Investigation of the impact of microbial agent treatment on controlling Astragalus membranaceus root rot disease[J/OL]. Chinese Journal of Pesticide Science, 1-14[2025-04-30]. https://doi.org/10.16801/j.issn.1008-7303.2025.0008.

[27] Yang, B., Li, Y.X., Xu, J.Y., et al. Screening, Identification, and Application Effects of Actinomycetes on Astragalus membranaceus Root Rot. [J/OL]. Chinese Journal of Biological Control, 1-7[2025-05-27]. https: //doi.org/10.16409/j.cnki.2095-039x.2025.02.024.

[28] Zhang, X.C., Zhang, H.J., Li, S.B., et al. The control effect of Trichoderma harzianum EMF910 on theroot rot pathogens of Astragalus membranaceus in Ningxia saline-alkali regions[J]. Microbiology China, 2024, 51(10): 4162-4180. DOI: 10.13344/j.microbiol.china.231041.

Comments 5: In conclusion, is it worth to describe in detail the advantage of this test in comparison with the other available? Will this method be a universal?

Response: Thanks for your acknowledgement for our study. Your comments were carefully addressed to improve the methodology rigorousness and writing readability.

We have added relevant content in conclusion. The in vitro control efficacy of fungicides and biocontrol agents against Fusarium oxysporum and Fusarium solani, the pathogenic fungi causing root rot in Astragalus membranaceus, was evaluated, resulting in the screening of highly effective fungicides and biocontrol strains. Carbendazim can be prioritized for field screening in Astragalus root rot control. The results could provide a foundation for future research on microbial biocontrol formulations.

Comments 6: The authors need to check the entire manuscript for grammar and spelling.

Response: Thanks for your acknowledgement for our study. Your comments were carefully addressed to improve the methodology rigorousness and writing readability. 

Reviewer 3 Report

The similarity percentage is acceptable at 14%. However, the document reveals several significant methodological gaps and conceptual errors. It lacks sufficient statistical analyses and presents averages with notably wide standard deviations, as shown in Table 2 of the Materials and Methods section. Moreover, it does not explain how the plant material was inoculated with the bacteria. The authors mistakenly conflate the culture media PDB with PDA and fail to provide evidence regarding the quality and integrity of the DNA, among other concerns.

no additional comments

Author Response

Comment 1:The similarity percentage is acceptable at 14%. However, the document reveals several significant methodological gaps and conceptual errors. It lacks sufficient statistical analyses and presents averages with notably wide standard deviations, as shown in Table 2 of the Materials and Methods section. 

Response: Thanks for your acknowledgement for our study. Your comments were carefully addressed to improve the methodology rigorousness and writing readability. The table 2 was in the results section. The experimental results in this article all have three or more replicates. The data sources are reliable. Due to the experimental procedures and sample characteristics, the standard deviation of the average values may be relatively large. We will improve these issues in future experiments.

Comment 2:Moreover, it does not explain how the plant material was inoculated with the bacteria.

Response: Thanks for your acknowledgement for our study. We have included detailed information on the pathogenic bacteria and the bacterial inoculation methods in the section about the antibacterial effects of bacteria on pathogenic organisms. As following: 5-mm diameter test pathogen discs were inoculated onto 90-mm diameter PDA plates. The biocontrol bacteria were streaked in a parallel diameter direction 20 mm away from the center of the plate. A control plate without inoculating the biocontrol bacteria was used as a reference. Each treatment was repeated 3 times. The plates were placed in a 25°C dark environment for 5 to 7 days. (Page 6, Lines 214-219)

Comment 3:The authors mistakenly conflate the culture media PDB with PDA and fail to provide evidence regarding the quality and integrity of the DNA, among other concerns.

Response: Thanks for your acknowledgement for our study. We have conducted checks and made corrections on the experiments using PDA and the agar-free PDA medium as described in the text.

Round 2

Reviewer 2 Report

Accept in present form.

This work is interesting and it meets the scope of the journal.

Author Response

Comments 1: This work is interesting and it meets the scope of the journal.

Response 1: Thanks for your acknowledgement for our study.

Reviewer 3 Report

The document continues to contain conceptual errors; the authors should clarify that there is no PDA without agar. PDA is the acronym for Potato Dextrose Agar, and PDB is the abbreviation for Potato Dextrose Broth, without agar.

The document continues to contain conceptual errors; the authors should clarify that there is no PDA without agar. PDA is the acronym for Potato Dextrose Agar, and PDB is the abbreviation for Potato Dextrose Broth, without agar.

Author Response

Comments 1: The document continues to contain conceptual errors; the authors should clarify that there is no PDA without agar. PDA is the acronym for Potato Dextrose Agar, and PDB is the abbreviation for Potato Dextrose Broth, without agar.

Response 1: Thanks for your acknowledgement for our study. Your comments were carefully addressed to improve the methodology rigorousness and writing readability. We have corrected the incorrect usage of PDA medium and PDB medium in the text. The revised version is as follows:

  1. Potato Dextrose Broth (PDB):Patato 200 g, Dextrose 20 g, distilled water 1000 ml. (Page 3, Lines 107-108)
  2. Then, the mycelium at the edge of the colonies was inoculated into PDB medium  and cultured at a temperature of 25℃ and 180 r/min for 5-7 days. (Page 4, Lines 137-139)
  3. After the mycelium covered the entire petri dish, a sterile punch with a diameter of 5 mm was used to cut out mycelial plugs, which were then inoculated into PDB medium and cultured on a shaker (28℃, 160 r/min) for 5 to 7 days. (Page 4, Lines 154-156)